# Trypanosomes of the *Trypanosoma theileri* Group: Phylogeny and New Potential Vectors

**DOI:** 10.3390/microorganisms10020294

**Published:** 2022-01-26

**Authors:** Anna Brotánková, Magdaléna Fialová, Ivan Čepička, Jana Brzoňová, Milena Svobodová

**Affiliations:** 1Department of Parasitology, Faculty of Science, Charles University, Vinicna 7, 128 44 Prague, Czech Republic; fialomag@natur.cuni.cz (M.F.); milena.svobodova@natur.cuni.cz (M.S.); 2Department of Zoology, Faculty of Science, Charles University, Vinicna 7, 128 44 Prague, Czech Republic; ivan.cepicka@natur.cuni.cz

**Keywords:** *Trypanosoma theileri*, *Trypanosoma melophagium*, mosquito, *Phlebotomus*, tabanid, ked, vector, phylogeny, prediuresis, transmission

## Abstract

Trypanosomes belonging to *Trypanosoma theileri* group are mammalian blood parasites with keds and horse fly vectors. Our aim is to study to vector specificity of *T. theileri* trypanosomes. During our bloodsucking Diptera survey, we found a surprisingly high prevalence of *T. theileri* trypanosomes in mosquitoes (154/4051). Using PCR and gut dissections, we detected trypanosomes of *T. theileri* group mainly in *Aedes* mosquitoes, with the highest prevalence in *Ae. excrucians* (22%), *Ae. punctor* (21%), and *Ae. cantans/annulipes* (10%). Moreover, *T. theileri* group were found in keds and blackflies, which were reported as potential vectors for the first time. The vectorial capacity was confirmed by experimental infections of *Ae. aegypti* using our isolates from mosquitoes; sand fly *Phlebotomus perniciosus* supported the development of trypanosomes as well. Infection rates were high in both vectors (47–91% in mosquitoes, 65% in sandflies). Furthermore, metacyclic stages of *T. theileri* trypanosomes were observed in the gut of infected vectors; these putative infectious forms were found in the urine of *Ae. aegypti* after a second bloodmeal. On the contrary, *Culex pipiens quinquefasciatus* was refractory to experimental infections. According to a phylogenetic analysis of the 18S rRNA gene, our trypanosomes belong into three lineages, TthI, ThII, and a lineage referred to as here a putative lineage TthIII. The TthI lineage is transmitted by Brachycera, while TthII and ThIII include trypanosomes from Nematocera. In conclusion, we show that *T. theileri* trypanosomes have a wide range of potential dipteran vectors, and mosquitoes and, possibly, sandflies serve as important vectors.

## 1. Introduction

Trypanosomes (Euglenozoa; Kinetoplastea; Trypanosomatida) [1] belong among the most important and widespread parasites worldwide, causing important diseases in humans and livestock. They are digenetic blood parasites transmitted mainly by various bloodsucking insects. Trypanosomes of the *Trypanosoma theileri* group (*T. theileri* henceforth) have been reported from various ungulates in cattle, buffaloes, sheep, antelopes, and deer [2,3,4,5,6,7]. Although widespread, the *T. theileri* group is largely neglected due to its low economic importance and causing no pathology [2,3]. Infections by *T. theileri* are mostly cryptic; however, pathologies might have resulted from coinfections or stress when fever, anorexia, and anemia were reported as symptoms in several bovid infections [8,9,10,11,12,13,14,15].

The *Trypanosoma theileri* group consists of several species (*Trypanosoma theileri*, *T. melophagium*, *T*. *cervi*, and *T. trinaperronei*) and various trypanosome genotypes reported from cervids, bovids, and insects [6,16,17,18,19]. Some genotypes are specific for a single host species, such as sheep or water buffalo [3,16,20,21], while others, belonging to both TthI and TthII lineages, have been reported from cattle and deer [3,16,17,22].

Mammals are infected by ingesting the vector with metacyclic trypomastigotes or by contamination of skin abrasion or mucous membrane by feces of the vector [2,23,24]. A possible transplacental transmission was considered in bovids and cervids [25,26]. Prediuresis (i.e., removing excess water to concentrate the bloodmeal) of the bloodfeeding vectors represents another potential transmission mode [27,28]. Prediuresis was described in various bloodsucking insects, such as kissing bugs, tsetse flies, sand flies, and mosquitoes [27,29,30,31,32,33,34]. Putative infectious stages of kinetoplastids were found in the urine of kissing bugs, sand flies, and mosquitoes [33,34,35]. Putative infectious stages of avian trypanosomes, probably belonging to the same *Megatrypanum* subgenus as *T. theileri* [36], were observed in mosquito urine [33].

Trypanosomes of *Trypanosoma theileri* group were detected in different groups of Diptera, such as tabanids [2,23,37,38], deer keds [6,39,40], mosquitoes [41], *Phlebotomus perfiliewi* [42], and tsetse flies [43,44]; in addition, they were also reported from several species of ticks: *Hyalomma anatolicum*, *Amblyomma americanum*, *Boophilus microplus,* and *Ornithodoros moubata* [45,46,47,48]. Despite deer keds having been assumed to be vectors of *T. theileri* trypanosomes [6,39], tabanids are the vectors confirmed by experimental infections [23], and development of *T. theileri* trypanosomes, including metacyclic stages, was described in the tabanid gut [37,38]. Furthermore, sheep ked *Melophagus ovinus* was confirmed as a vector of *T. melophagium*, a species belonging to the *T. theileri* group but occurring exclusively in sheep [2,49,50]. 

Mosquitoes are not considered important vectors of trypanosomes, and only a few studies have focused on them. *Culex* mosquitoes are confirmed vectors of the bird species *Trypanosoma culicavium* and *T. thomasbancrofti* [33,51]. The role of mosquitoes in the lifecycle of *T. theileri* is unclear; this trypanosome was detected in seven mosquito species but only using PCR [41].

The two main lineages of the *T. theileri* group, TthI and TthII, were previously defined based on analyses of ITS (internal transcribed spacer), and SL (spliced leader) genes [16,52]. Subsequently, several trypanosome strains were placed into TthI or TthII based on 18S rRNA gene phylogenies [17,38,53,54]. Recently, an additional lineage was distinguished based on 18S rRNA gene analysis [6].

During studies of trypanosome vectors, we have found a surprisingly high prevalence of *T. theileri* trypanosomes in examined mosquitoes by dissections and PCR. The infections in dissected mosquitoes suggested possible vectorial capacity. Therefore, we decided to focus on mosquitoes as possible vectors of *T. theileri* by examining wild mosquitoes and using *T. theileri* isolates for experimental infections of possible vectors. We also carried out a phylogenetic analysis of *T. theileri* based on the 18S rRNA gene sequences to assess the associations of *T. theileri* with various vectors.

## 2. Materials and Methods

### 2.1. Insect Trapping and Processing, and Sampling of Deer Blood

Mosquitoes were trapped monthly from May to August in 2017–2019 in three localities in the Czech Republic, namely Choteč (49.9991 N, 14.2802 E), Zeměchy (50.2318 N, 14.272 E), and Milovice forest (48.8213 N, 16.6932 E). Six CDC light traps (JW, Hock Company, Gainesville, FL, USA) baited with dry ice were used at each trapping event. Traps were installed between 4:00 p.m. and 6:00 p.m. and removed between 8:00 and 10:00 a.m. the next day. Collected insects were killed in a −80 °C freezer or a box with dry ice and were sorted by families. Mosquitoes were stored in Petri dishes at −20 °C for species determination.

Tabanids were collected in the Milovice forest in 2019, mainly as bycatch in mistnets; some were caught in CDC traps (see above) or by hand inside a car.

In 2017–2018, sheep keds *Melophagus ovinus* were collected from sheep at six localities: Vlkov (49.1512 N, 14.7252 E), Ratíškovice (48.9200 N, 17.1656 E), Hořice (50.3661 N, 15.6318 E), Statenice (50.1426 N, 14.3185 E), Valašská Senice (49.2253 N, 18.1169 E), and Přerov Předmostí (49.4675 N, 17.4374 E). Deer keds were collected by hunters in 2017–2019 directly from shot fallow deer (*Dama dama*), red deer (*Cervus elaphus*), and roe deer (*Capreolus capreolus*) at Boršov nad Vltavou (48.9218 N, 14.4340 E), Milovice forest (48.8213 N, 16.6932 E), Blíževedly (50.6084 N, 14.3965 E), Planá (49.8682 N, 12.7438 E), Bystřice (49.7321 N, 14.6674 E), Neveklov (49.7537 N, 14.5329 E), Nové Strašecí (50.1527 N, 13.9004 E), Obecnice (49.7162 N, 13.9473 E), and Vonoklasy (49.9501 N, 14.2767 E). Blood samples were collected by hunters from the shot game in Blíževedly (50.6084 N, 14.3965 E), Doupov (50.2572 N, 13.1432 E), Hvězda (50.6023 N, 14.4396 E), Kublov (49.9437 N, 13.8767 E), Litice (50.6134 N, 14.4393 E), Milovice forest (48.8213 N, 16.6932 E), Nižbor (49.1000 N, 14.0024 E), Nové Hrady (48.7896 N, 14.7783 E), Nové Strašecí (50.1527 N, 13.9004 E), Skalka (50.5857 N, 14.4118 E), and Vonoklasy (49.9501 N, 14.2767 E). Keds for dissection were stored alive in zip-lock bags; dead keds were stored in ethanol. A sample of game blood was fixed in ethanol for PCR detection, and another was used for cultivation (see below).

The insects were identified under a stereomicroscope using determination keys [55,56]; undetermined insects were barcoded when possible [57]. Insects (except tabanids) were pooled according to the species and locality in pools containing ten or fewer specimens and were examined using nested PCR (see below). Engorged insects with visible blood in the gut were processed individually. Some living keds, tabanids, and mosquitoes were killed and dissected, and their intestines were examined for the presence of trypanosomes.

### 2.2. Dissection and Cultivation

Insects were killed and washed in 70% ethanol, followed by a sterile saline solution. The gut was dissected in a drop of sterile saline under a stereomicroscope, and infection status was checked under a light microscope. Parasite localization, appearance, and quantity were recorded. Infections were considered to be weak if fewer than 100 parasite cells were visible, moderate with 100–1000 cells present, and heavy with more than 1000 cells per gut. A part of positive guts was used to cultivate kinetoplastids, and the rest was stored in ethanol for PCR detection. 

Kinetoplastids from positive guts and deer blood samples were cultivated in 4 mL glass vials on rabbit (Bioveta, Ivanovice na Hané, Czech Republic) or sheep (LabMediaServis, Jaroměř, Czech Republic) blood agar (SNB-9) overlayed with RPMI 1640 (Sigma–Aldrich, St. Louis, MO, USA) and Schneider Drosophila Medium (Sigma–Aldrich, St. Louis, MO, USA) in a 1:1 ratio supplemented with 20% FCS (Gibco, Thermo Fisher Scientific, Inc., Waltham, MA, USA), 2% sterile human urine, 100 µg/mL amikacin (Medochemie, Prague, Czech Republic), 5000 U/mL penicillin, and 1.5 mg/mL 5-fluorocytosine (Sigma–Aldrich, St. Louis, MO, USA) at 23 °C. The presence of kinetoplastids was checked weekly. Thriving cultures were subcultured into flat tubes with blood agar and cryopreserved in liquid nitrogen. Trypanosomes for experimental infections were cultivated in the same medium without fluorocytosine and penicillin.

### 2.3. PCR Detection of Kinetoplastids, Sequencing

DNA was extracted using the High Pure PCR Template Preparation Kit (Roche Diagnostic, Manheim, Germany) according to the manufacturer’s instructions. EmeraldAmpGT PCR Master Mix (TaKaRa Bio, Kusatsu, Shiga, Japan) was used for PCR reactions. The 18S rRNA gene was amplified using a single-step or nested PCR. MedA (CTGGTTGATCCTGCCAG) and MedB (TGATCCTTCTGCAGGTCCACCTAC) primers [58] were used to amplify the DNA from cultures obtained from the positive dissected insect guts or deer blood samples. Conditions were as follows: denaturation temperature was 94 °C for 5 min followed by 30 cycles at 94 °C for 1 min, 55 °C for 1 min 30 s, 72 °C for 1 min 30 s, and final extension at 72 °C for 5 min. Nested PCR was used to detect kinetoplastids in the dead insects, positive guts, and deer blood. Primers S762 (GACTTTTGCTTCCTCTAWTG) and S763 (CATATGCTTGTTTCAAGGAC) [59] were used for the first step with the same cycle condition as single-step PCR. TRnF2 (GARTCTGCGCATGGCTCATTACATCAGA) and TRnR2 (CRCAGTTTGATGAGCTGCGCCT) primers [43] were used for the second step with the same conditions as the single-step PCR but with an annealing temperature of 64 °C.

PCR products of the positive samples (visualized in gel electrophoresis) were purified by ExoSAP (Thermo Fisher Scientific, Inc., Waltham, MA, USA) according to the manufacturer’s instructions. Primers 1000R (ATGCCTTCGCTGTAGTTCGTCT) and 1000F (AGACGAACTACAGCGAAGGCAT) [60] and 577F (GCCAGCACCCGCGGT) [61] were used for sequencing.

### 2.4. Prevalence of T. Theileri

The prevalence was calculated as the Minimal Infection Rate (MIR) as follows:$$\mathrm{MIR}\left(\%\right)=\frac{n\;\mathrm{of}\;T.\;theileri\;\mathrm{positive}\;\mathrm{pools}}{n\;\mathrm{of}\;\mathrm{examined}\;\mathrm{pools}}\cdot 100$$

Prevalence was counted when at least 15 individuals were examined.

### 2.5. Experimental Infections and Prediuresis Experiments

Mosquitoes *Aedes aegypti*, *Culex pipiens molestus, Cx. p. quinquefasciatus,* and the sandfly *Phlebotomus perniciosus* were used for experimental infections. *Culex* and *Phlebotomus* were permanently reared at the Department of Parasitology, Charles University, Prague, Czech Republic. A colony of *Ae. aegypti* was temporarily established; mosquitoes were obtained from The National Institute of Public Health, Czech Republic. Colonies were maintained at 25 °C and 80% relative humidity. About 100 females were exposed to parasites by feeding through a chick skin membrane on heat-inactivated rabbit or sheep blood (30 min at 56 °C) containing 5–7 days old culture of 10^7^ parasite cells/mL. Due to autogeny of *Cx. p. molestus*, fed mosquitoes were sorted after feeding. In other species, fed specimens were recognized during dissection by the presence of developing eggs. Ambient humidity and 50% sucrose solution on a cotton pad were provided to blood-fed insects. All trypanosome isolates used in the experiments were our own and are summarized together with temperature conditions in Table 1. Low temperatures were used to mimic the natural conditions, as some kinetoplastids are known to develop better at lower temperatures [62]. After defecation (10–62 post-infection for mosquitoes, 7–17 post-infection for sandflies), guts were dissected at several time points and examined under a light microscope for infection status, infection intensity, and parasite localization. 

*Aedes aegypti* mosquitoes experimentally infected with the *Trypanosoma* isolate CUL46 were used for the prediuresis experiment 22 days after infection. Mosquitoes were blood-fed through a membrane, and, immediately after feeding, they were placed individually in tubes with coverslips at the bottom. After defecation, the coverslips were dried, fixed with methanol, and stained with Giemsa–Romanowski (Sigma–Aldrich, St. Louis, MO, USA). 

### 2.6. Light and Scanning Electron Microscopy

Dissected positive mosquito guts and samples from prediuresis were fixed on slides with methanol and stained with Giemsa–Romanowski. Slides were examined under the light microscope Olympus BX51 TF with a CDC camera (DP70), and cells were photographed with software QuickPHOTO CAMERA 3.2. ImageJ software was used for the measuring of cell length [63]. Positive guts of *Ae. aegypti* and *Ph. perniciosus* from experimental infections were prepared for scanning electron microscopy (JEOL 6380 LV) as described earlier [33]. 

### 2.7. Phylogenetic Analysis

A dataset of the 18S rRNA gene sequences consisted of 238 *T. theileri* sequences from mosquitoes, tabanids, black flies, deer keds, sheep keds, and deer blood. *T. avium* (KT728402), *T. grayi* (KF546526), *T. microti* (AJ009158), and *T. conorhini* (AJ012411) were used as an outgroup. The sequences were aligned by MAFFT [64] with the MAFFT server (https://mafft.cbrc.jp/alignment/server/, accessed on 24 January 2022) and the following algorithms and parameters: G-INS-I, 200PAM/κ = 2, the penalty for the first gap 1.53, offset value 0.0 and N does not affect the alignment score. BioEdit 7.2.5 [65] was used for manual alignment masking. The final dataset consisted of 1800 positions. RAxML 8.2.10 [66] with the GTRGAMMAI model was used for a maximum-likelihood analysis, which was conducted with 100 repeated tree searches. The tree was bootstrapped with 1000 replicates. 

## 3. Results

### 3.1. Prevalence of T. theileri in Mosquitoes

A total of 4051 mosquito females belonging to 18 species were caught in 2017–2019; from these, 3282 were tested by PCR in 560 pools, and 769 specimens were examined by dissection of the gut. The most abundant species were *Cx. pipiens, Ae. vexans*, and *Mansonia richiardii*. *T. theileri* was detected in 14 mosquito species belonging to five genera (*Aedes, Anopheles, Culex, Culiseta*, and *Mansonia*). The prevalence ranged from 0.05% in *Cx. pipiens* to 21.7% in *Ae. excrucians* (Figure 1). Findings of *T. theileri* in minority species include *Ae. cataphylla* (1/2), *Ae. sticticus* (1/2), *An. claviger* (1/6), and *An. plumbeus* (2/13). Four tested mosquito species were *T. theileri* negative, *Ae. caspius* (*n* = 40), *Ae. flavescens* (*n* = 4), *Cx. modestus* (*n* = 8) and *Cs. morsitans* (*n* = 2).

### 3.2. Prevalence of T. theileri in Deer Keds

Three *T. theileri* positive specimens of *Lipoptena fortisetosa* were detected among 224 examined (Appendix A). No trypanosomes were detected in *L. cervi* (*n* = 22) and *L*. sp. (*n* = 2). Positive keds originated from two red deer (Milovice forest) and a roe deer (Bystřice). MIR is 1.2% (3/248).

### 3.3. Prevalence of T. melophagium in Sheep Keds

By PCR, *T. melophagium* was detected in 53 from 79 tested pools (67%, Hořice), and three *T. melophagium* isolates (MOVI1–3) were obtained from sheep keds from Vlkov. A total of 184 sheep keds were examined, and the prevalence of 33% was calculated per site to prevent pseudoreplication when the keds came from the same sheep or herd. For numbers of sheep keds from individual localities, see Appendix A.

### 3.4. Prevalence of T. theileri in Tabanids

Twenty-five tabanids of four species (*Hybomitra ciureai, Tabanus bromius, Haematopota pluvialis*, and *Atylotus leowianus*) were caught in the Milovice forest. Fifteen tabanids (60%) were positive for kinetoplastids by dissection, and subsequent sequencing confirmed *T. theileri* in 11 individuals with the prevalence of 44% (Table 2).

### 3.5. Comparison of T. theileri Prevalence in Insects

The highest *T. theileri* prevalence of 44% was found in tabanids. *T. theileri* prevalence (18%) in deer keds is counted per mammalian host to prevent pseudoreplication. Overall, in *Aedes* mosquitoes, a prevalence of 7% was counted, and only 1% of blackflies were positive for *T. theileri* (Figure 2). 

### 3.6. Detection of T. theileri in Deer Blood

We collected 33 deer blood samples from red deer (*n* = 24), roe deer (*n* = 7), and fallow deer (*n* = 2). All samples were PCR negative. However, two out of five samples tested by cultivation were positive for *T. theileri* from red deer in Doupov (CELA1) and Milovice forest (CELA2).

### 3.7. Experimental Infections of Mosquitoes

#### 3.7.1. Experiments with Cx. *p. quinquefasciatus* and Cx. *p*. *molestus*


Trypanosomes failed to develop in 95 *Culex* specimens kept at 21 °C (Appendix A); weak (*n* = 2) and moderate (*n* = 4) infections were detected in *Culex* mosquitoes kept at 15 °C (*n* = 87) or transferred from initial temperature of 8–11 °C to 15 °C (*n* = 20). Undigested blood was still observed in the gut of several dissected mosquitoes kept at 8–11 °C after the 21st day. Therefore, only defecated mosquitoes were considered positive. Weak (*n* = 6) to moderate (*n* = 4) infections were detected in *Cx. p. quinquefasciatus* (*n* = 52). In 29 tested *Cx. p. molestus*, moderate (*n* = 1) and weak (*n* = 5) infections were found (Figure 3). Moderate infections were localized in the abdominal midgut or hindgut; trypanosomes were in rosettes or present as individual cells. Weak infections were localized in the abdominal midgut.

#### 3.7.2. Infectious Experiments with *Aedes aegypti*

*Aedes aegypti* was successfully infected with strains isolated from mosquitoes (*Cs. annulata* and *Ae. vexans*), with prevalence ranging from 47% to 91% (Figure 3). Most of the mosquitoes were heavily infected in both experiments. Trypanosomes formed immotile rosettes localized primarily in the area of the rectal ampulla, but in the case of very heavy infection, they were also found in other parts of the hindgut. Free cells of trypanosomes were round or pear-shaped, not very mobile, and seemed to be aflagellated under the light microscope. Giemsa-stained positive mosquito guts revealed a presence of epimastigotes, sphaeromastigotes, and metacyclic stages. Guts with heavy infection were used for scanning electron microscopy. The parasites were observed with hemidesmosome attached to the intestine wall (Figure 4c,d).

The susceptibility of mosquitoes was low for the isolate TAB1 (ex *H. ciureai*), with a prevalence of 6%. Two moderate and two weak infections were localized in the rectum. Rosettes were present in one case only. In addition, both motile and immotile unattached parasites were observed.

All mosquitoes experimentally fed on the isolates MOVI1 (ex *M. ovinus*; *n* = 36) and CELA1 (ex *C. elaphus*; *n* = 50 and *n* = 38) were negative.

#### 3.7.3. Experimental Infection of the Sand Fly *Phlebotomus perniciosus*

The prevalence of 65% was detected in *T. theileri*-infected sandflies, and most positive females had heavy infections (Figure 3). Free cells of trypanosomes were noticed in various parts of the gut (rectum, hindgut, abdominal midgut) in weak infections, and rosettes were observed in heavy infections, similar to experiments with *Ae. aegypti.* Moderate and heavy infections were localized in the hindgut, mainly in the rectum. Two types of cells were seen under the light microscope: moving epimastigotes (Figure 4a) and rounded, aflagellated cells with minimal motility. Giemsa-stained positive gut showed a presence of epimastigotes, sphaeromastigotes, and metacyclic stages (Figure 5). Scanning electron microscopy revealed trypanosomes with hemidesmosomes as in *Ae. aegypti* guts (Figure 4b).

### 3.8. Morphology of Trypanosomes in Vectors

Epimastigotes and metacyclic stages were observed and measured in the guts of positive tabanids (Table 3). Elongated or droplet-shaped epimastigotes were identified in the abdominal midgut and hindgut of tabanids.

*Trypanosomes* originating from the experimental infection of *Ae. aegypti* or *Ph. pernicious* with the strain CUL46 were also measured (Table 3). Elongated and droplet-shaped epimastigotes, spheromastigotes, and infectious stages were observed (Figure 5a–d). The flagella were not always seen in epimastigotes. Some epimastigotes were observed in rosettes (Figure 5e).

*Trypanosoma theileri* epimastigotes, sphaeromastigotes, and metacyclic stages were observed in six out of 18 coverslips in prediuresis experiments (*Ae. aegypti*, CUL46) (Figure 5f,g). The length of metacyclic stages ranged from 3.1 µm to 6.6 µm, and the average was 5.3 µm.

### 3.9. Phylogenetic Analysis

The phylogenetic tree of the *Trypanosoma theileri* group as inferred from the 18S rRNA gene is shown in Figure 6. In our 18S rRNA gene tree, TthI was recovered paraphyletic. TthII appeared monophyletic, though with low support (bootstrap support, BS, 51). Sequences AY971802 and AY971803 from tabanids, which were previously placed into the lineage TthII solely based on 18S rRNA gene [6], were not closely related to TthII in our tree (Figure 6, marked with †). We, therefore, do not consider them to belong to TthII. Most of our newly determined sequences (130 out of 170) were placed into the lineage TthII. Some of them were identical or nearly identical to already published sequences, but two novel groups within TthII, mainly consisting of sequences obtained from mosquitoes, were identified (box in Figure 6). Seven new sequences clustered with TthI. The remaining 33 sequences formed a robust clade (BS 99), which was distinct from TthI and TthII; It is here referred to as the putative *T. theileri*-group lineage TthIII. Besides the newly determined sequences from mosquitoes, TthIII contained two previously published sequences from deer [22]. TthIII further split into two clades (BS 86 and 65, respectively). 

## 4. Discussion

*Trypanosoma theileri* has been recently detected in several potential vector groups (keds, mosquitoes, and a sandfly). However, the relevance of these findings is still unclear since the lifecycle was not confirmed, and metacyclic stages were reported only in tabanids (*T. theileri*) [23,67] and sheep keds (*T. melophagium*) [2,49,50]. Our study focused not only on the molecular detection of *T. theileri* in bloodfeeding insects and mammalian hosts but also on its development and infectious stages occurrence in the intestine of potential vectors to assess the lifecycle and host/vector range of this species. We detected *T. theileri* in various bloodsucking Diptera, and it is obvious that it is not only tabanids and keds that can transmit these trypanosomes. 

Similar to previous studies, we have detected *T. theileri* lineage TthI and TthII; in addition, a third lineage, whose existence was revealed previously [6], was here designated as TthIII. Some mosquito *T. theileri* sequences created separate groups in the TthII and TthIII lineages. Furthermore, we revealed different vectorial specificity of lineages since Brachycera transmit the TthI while other lineages are transmitted by both Brachycera and Nematocera. 

Several mosquito species were infected with *T. theileri,* some of which had a high prevalence (22% in *Aedes excrucians*). *Aedes* mosquitoes are considered opportunistic or mammalophilic [68,69]; the abundance of mammals is high in some of the studied localities (game reserve Milovice forest), enabling intensive circulation of the parasite. Contrary to the genus *Aedes*, *Culex* mosquitoes are considered ornithophilic [70]; the low prevalence of *T. theileri* in this genus is thus not surprising. Nevertheless, a single positive specimen confirms the willingness of *Culex* to feed on mammals reported previously [41,70]. The detected prevalence of 7% in *Aedes* mosquitoes (including nulliparous females) and heavy, mature infections in naturally infected mosquitoes suggests that they are effective vectors of *T. theileri*. 

*Trypanosoma melophagium* was isolated from sheep keds with a prevalence of 67% in Hořice, similar to a previous study from Scotland [20]. The high prevalence is influenced by the fact that sheep keds do not leave their host and suck blood daily [71], so the probability of being positive is high when a sheep is infected [21].

The keds *Lipoptena cervi* and *L. fortisetosa* were collected from the cervids. In *L. cervi,* we did not detect *T. theileri* trypanosomes, possibly due to a small number of tested keds. However, *T. theileri* was found in *L. fortisetosa*. *T. theileri* was detected by PCR in both deer ked species in Poland [40]. However, PCR alone is not sufficient to confirm the transmission potential of a positive vector [72], and we did not find any infection in dissected keds. Therefore, it remains unclear if *L. fortisetosa* is a vector. Böse and Petersen [39] described rosettes and epimastigotes in the intestine of *L. cervi* but did not report any metacyclic stages; they did not perform transmission experiments either. Similarly, *L. mazamae* was predicted as a possible vector of the recently described *Trypanosoma trinaperronei* [6], which was detected in this ked species by PCR, but neither development in the intestine nor metacyclic stages were described in the study. After finding a host, deer keds drop their wings, limiting their potential to switch hosts. However, the exchange of keds among animals in a herd by direct contact is possible, although not to the same extent as in sheep herds and *M. ovinus*, where the animals are in close contact [71]. Moreover, a recent study has detected trypanosomes in unfed, winged deer keds [40]. This observation begs a hypothesis of *T. theileri* transmission from adult females to larvae through feeding glands in keds [40]. *Lipoptena* spp. could have a role as additional vectors. Since keds feed on their host frequently, and blood is present in their gut permanently [56], PCR positivity does not necessarily prove keds as specific vectors of *Trypanosoma theileri* [21,72].

Tabanids such as *Tabanus bromius* have been previously confirmed as *T. theileri* vectors of cattle and deer trypanosomes [23]. The high prevalence detected during our study (44%) is slightly higher than those detected in Poland (34%) or Russia (31%) [38,73] and suggests a significant role of tabanids in *T. theileri* transmission at these study sites. We also record *T. theileri* in *H. ciureai* for the first time. 

This is the first record of *T. theileri* in blackflies where the bloodmeal was detected in only one specimen out of nine positive. Black flies thus could have an additive role in transmission, but the prevalence was low (1%).

In the case of vertebrate host blood, *T. theileri* was detected using cultivation in two samples, while PCR gave negative results in all 33 specimens of blood. Blood cultivation seems to be a more sensitive diagnostic method; however, it is prone to contamination with yeast and bacteria [74,75], especially when using blood from shot animals. 

Experiments with laboratory-bred vectors correspond with field observations. *Ae. aegypti* was highly sensitive to *T. theileri* clade II infection with 67% infected specimens and 97% of heavy infections, and the occurrence of metacyclic stages, identical to those previously described from tabanids and sheep keds [2,37,38,49,50]. Furthermore, in the infected gut, the length of epimastigotes (12.7–23.5 μm) corresponded to the earlier observation of epimastigotes that proliferate into shorter cells [2,37]. Both the observed infection intensity and cell morphology, therefore, support our conclusion that *Aedes* mosquitoes are competent vectors. Most importantly, *T. theileri* metacyclic stages were detected in the urine of infected mosquitoes during our prediuresis experiments. It has been confirmed experimentally that mammalian trypanosomes can be transmitted through conjunctiva [34]. In conclusion, *Aedes* mosquitoes can be considered vectors of *T. theileri* clade II trypanosomes.

On the other hand, the role of *Culex* mosquitoes as vectors is likely negligible. In the infectious experiments, *Culex* mosquitoes were not infected with *T. theileri* at 21 °C or 15 °C, but a few moderate infections developed at 8–11 °C, which slowed down blood digestion, causing delayed defecation. In wild mosquitoes, only one out of 2128 tested *Culex* mosquitoes harbored *T. theileri*. Nevertheless, this infection was mature, without blood in the dissected gut, opening the potential of *Culex* mosquitoes as bridging vectors. Moreover, experiments with different environmental temperatures, which affect pathogen development, attachment, or invasion of the gut wall and influence transmission to a new host [62,76,77,78], showed a few weak and moderate infections only at a lower temperature. 

The infection experiments confirmed some extent of vector specificity among different *T. theileri* genotypes. Infections of laboratory-bred *Aedes aegypti* have been successful only using mosquito isolates (CUL46 *Culiseta annulata*, CUL107 *Aedes vexans*). Experimental infections with a sheep ked isolate were unsuccessful, agreeing with sheep keds as specific vectors of *T. melophagium*, and only a few moderate/weak infections were observed using a tabanid isolate. Negative results were obtained with the deer isolate CELA1, which, unlike the cultures obtained from the insects, belongs to the TthI clade. 

Interestingly, a sandfly species, *Ph. perniciosus,* was successfully infected by our mosquito isolate belonging to the *T. theileri* TthII lineage. Heavy infections were observed in the hindgut, similar to the finding of a single naturally infected specimen of *Ph. perfiliewi,* which belongs to the same subgenus, *Larroussius* [42]. In addition, short epimastigotes, sphaeromastigotes, and metacyclic stages were observed. These results suggest that phlebotomine sandflies could serve as additional vectors of the *T. theileri* TthII lineage. 

Analysis showed that our sequences belong to both the lineages, and a part of them belong to the putative TthIII lineage. Most of our sequences differed from the sequences available in GenBank. The TthI lineage mainly consists of sequences originating from bovids, cervids, and tabanids; these putative vectors were sampled in Brazil and Russia [16,38]. Besides finding the TthI lineage in one specimen of a horse fly *H. ciureai*, we found it in one deer ked *L. fortisetosa* and four *Aedes* mosquitoes. Mosquitoes harboring *T. theileri* TthI lineage contained undigested blood, which suggests that they are likely not the specific vectors. Based on the available data, the *T. theileri* TthI lineage is probably transmitted by Brachycera, since most sequences of vectors originate from tabanids.

On the other hand, the TthII lineage included all tested mammalophilic Diptera, including the sequence obtained from a sand fly and sequences from blackflies, and newly identified potential vectors. Since only one out of the nine examined blackflies had blood in the intestine, their vectorial role is highly probable. Furthermore, *T. theileri* TthII trypanosome was previously found in five out of 4512 screened specimens of biting midges (*Culicoides obsoletus, C. pulicaris, C. punctatus*) [79], which supports wide vectorial specificity that has been postulated previously [38]. However, most of the vector sequences were obtained from mosquitoes that belonged to the genera *Anopheles*, *Culiseta*, *Mansonia*, and, most frequently, *Aedes*. Mosquitoes thus represent a substantial part of the vectors. 

Experimental infections of mosquitoes further support the results obtained during field sampling. Strains of one TthII genotype, both isolated from mosquitoes, gave high infection rates and intensities in laboratory-bred vectors (mosquitoes and sand flies). Nevertheless, the development of other strains was not supported; namely TthI isolate CELA1 from deer, and TthII isolate MOVI1 from *T. melophagium* and TAB1 from *H. ciureai* were not infectious for mosquitoes. A possibility thus still exists that there is some extent of vectorial specificity among the genotypes of TthII clade. 

The results have shown a very high diversity of vectors of the *T. theileri* group. *Aedes* mosquitoes probably play a crucial role in transmitting some genotypes of *T. theileri*-related trypanosomes.

## 5. Conclusions

We conclude that mosquitoes of the genus *Aedes* are competent vectors of *T. theileri* TthII and putative TthIII trypanosome groups. Infection probably occurs by vector ingestion or prediuresis. Phlebotomine sandflies have the potential to serve as additional vectors. Mosquitoes host diverse *T. theileri* TthII lineages; TthI lineages are transmitted by bloodsucking Brachycera, while *T. theileri* trypanosomes from TthII have a wide variety of bloodsucking vectors in Diptera.

## Figures and Tables

**Figure 1 microorganisms-10-00294-f001:**
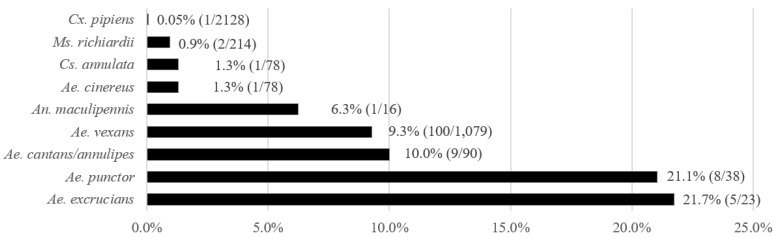
MIR of *T. theileri* for individual species with at least 15 examined individuals. Mosquito species are ordered by prevalence.

**Figure 2 microorganisms-10-00294-f002:**
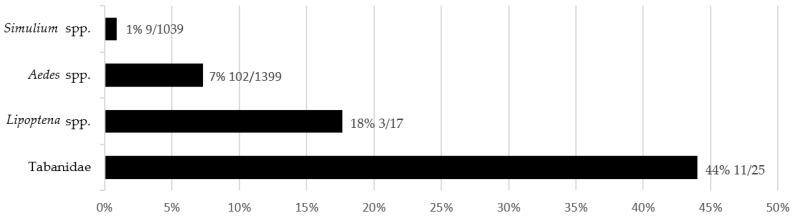
Comparison of *T. theileri* prevalence in bloodsucking insects. The prevalence of mosquitoes, tabanids, and blackflies is counted for insects trapped in the Milovice forest. The prevalence in deer keds was counted for insects collected at multiple localities.

**Figure 3 microorganisms-10-00294-f003:**
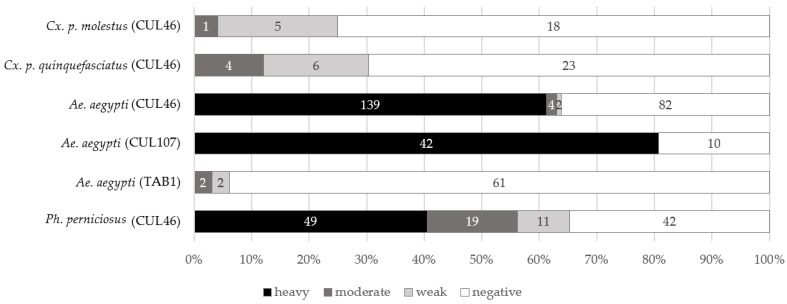
*T. theileri* prevalence in experimentally infected mosquitoes and sandflies. Prevalence for *Culex* spp. is shown for low-temperature experiments only.

**Figure 4 microorganisms-10-00294-f004:**
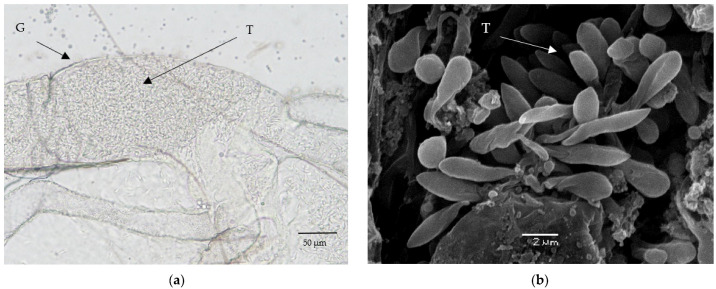
Light and electron microscopy. *T. theileri* in experimentally infected *Ph. perniciosus* (**a**,**b**) and *Ae. aegypti* (**c**,**d**). G—gut, T—trypanosomes in the disrupted gut, H—hemidesmosome.

**Figure 5 microorganisms-10-00294-f005:**
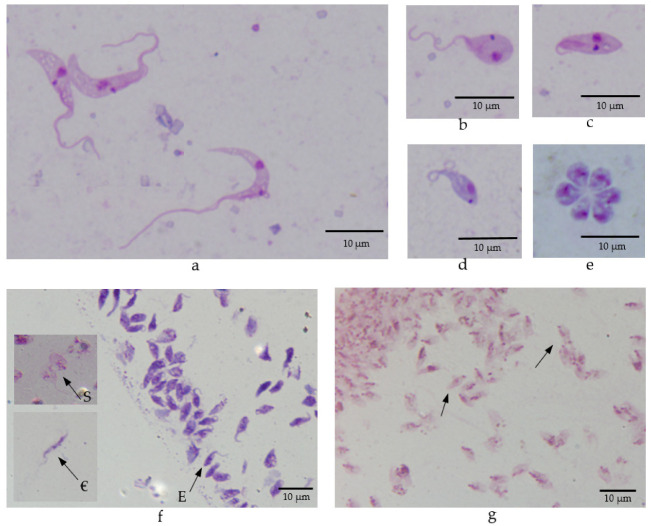
*T. theileri* morphotypes observed after infection of *Ph. perniciosus* (**a**–**d**) or *Ae. aegypti* (**e**) with CUL46 strain (ex *Cs. annulata*): **a**—elongated epimastigotes, **b**—spheromastigote, **c**—droplet-shaped epimastigote, **d**—metacyclic stage, **e**—rosette; (**f**,**g**): *T. theileri* morphotypes in prediuresis experiments with *Ae. aegypti* and CUL46 strain: **f**—elongated epimastigote (inset) (€), droplet-shaped epimastigotes (E), and spheromastigote (inset) (S), **g**—metacyclic stages (trypomastigotes; arrows).

**Figure 6 microorganisms-10-00294-f006:**
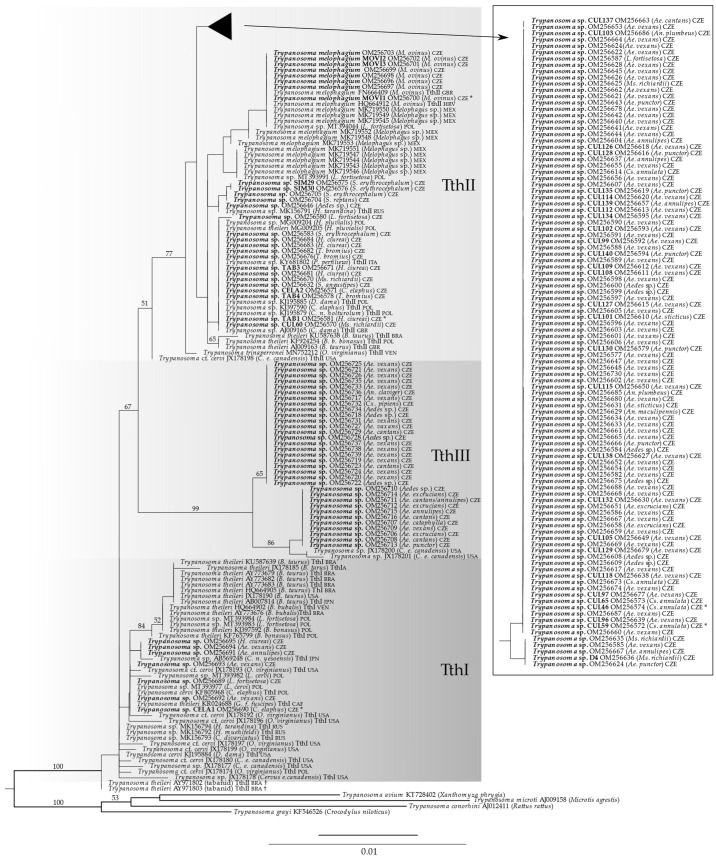
Phylogenetic tree of the *Trypanosoma theileri* group based on the 18S rRNA gene analysis. The tree was constructed using the maximum-likelihood method in RAxML (GTRGAMMAI model). Bootstrap values are shown at nodes. Newly determined sequences in bold. Cultures used in the infectious experiments are marked with asterisks. Daggers mark sequences which we do not consider to belong to TthII. The host species name is given in parentheses, followed by reported genotypes and abbreviations of the country of origin.

**Table 1 microorganisms-10-00294-t001:** Vector species, trypanosome isolates, and environmental temperatures used in the infectious experiments. 8–11→15: Fed mosquitoes were stored in fluctuating temperatures (8–11 °C), and after the 21st day, they were held at 15 °C.

Vector Species	Trypanosome Isolate	Environmental Temperature (°C)
*Culex p. quinquefasciatus*	CUL59 (CUL/CZ/2015/CUL59) ex *Culiseta annulata **	15
		21
*Cx. p. quinquefasciatus* *Cx. p. molestus*	CUL46 (CUL/CZ/2014/CUL46) ex *Cs. annulata **	8–11
		15
		8–11→15
		21
*Aedes aegypti*	CUL46 (CUL/CZ/2014/CUL46) ex *Cs. annulata*	21
	CUL107 (AED/CZ/2018/CUL107) ex *Aedes vexans*	
	CELA1 (CER/CZ/2017/CELA1) ex *Cervus elaphus*	
	TAB1 (HYB/CZ/2019/TAB1) ex *Hybomitra ciureai*	
	MOVI1 (MEL/CZ/2017/MOVI1) ex *Melophagus ovinus*	
*Phlebotomus pernicious*	CUL46 (CUL/CZ/2014/CUL46) ex *Cs. annulata **	21

* Strains obtained during studies of trypanosome vectors in previous years.

**Table 2 microorganisms-10-00294-t002:** *T. theileri* detection in tabanids. *n* specimens: number of tested samples, *n* Kinetoplastid+: number of kinetoplastid-positive specimens, *n T. theileri*+: number of *T. theileri* positive samples confirmed by sequencing.

Species	*n* Specimens	*n* Kinetoplastid + (Prevalence)	*n T. Theileri* + (Prevalence)
*Hybomitra ciureai*	16	11 (69%)	8 (50%)
*Tabanus bromius*	6	4	3
*Haematopota pluvialis*	2	0	0
*Atylotus leowianus*	1	0	0
Total	25	15 (60%)	11 (44%)

**Table 3 microorganisms-10-00294-t003:** Summary table of the measured length of *T. theileri*.

Morphotype	Tabanid Mean (Range) (µm)	Mosquito Mean (Range) (µm)	Sandfly Mean (Range) (µm)	Prediuresis Mean (Range) (µm)
Elongated epimastigote	16.0 (8.9–22.6) *	15.3 (12.6–23.5) *	16.8 (11.4–25.0)	-
Sphaeromastigote	-	8.4 (7.0–10.0) *	7.4 (3.6–13.9)	-
Droplet-shaped epimastigote	7.5 (5.1–11.0)	7.2 (3.3–23.0)	8.8 (5.0–16.5)	5.3 (4.3–7.0)
Metacyclic stages	4.5 (3.9–7.8)	5.0 (3.3–8.0)	5.5 (3.5–7.4)	5.3 (3.8–6.8)

* Less than 15 measured cells.

## Data Availability

The sequence data presented in this study are deposited in Genbank with accession numbers: OM256570-OM256739.

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
