# Peer review of "Trypanosomes of the Trypanosoma theileri Group: Phylogeny and New Potential Vectors"

_microorganisms, 2022, doi:10.3390/microorganisms10020294_

Round 1

Reviewer 1 Report

This study evaluates active transmission of T. theileri in select prevalent areas. The findings are important as investigators were able to detect mosquitoes as potential vector. These findings can led in successful planning and excecution of public health policies for disease control. 

Author Response

Thank you for your review.

We have corrected the missing data according to the reviewers´ suggestions.

Reviewer 2 Report

The manuscript clearly describes the vectorial capacity of Trypanosoma theileri. The authors determined that Aedes mosquitoes are potential vectors of T. heileri, also sand fly (Phlebotomus pernicious) supported development. The authors also investigated the prevalence in wild insects. The manuscript is well written and organized.

However, I addressed some issues below in my comments.

Line 108-109. Authors wrote that collected keds and blood were stored in ethanol and to detect T. theileri they used cultivation. What percentage of ethanol was used? And can material fixation in ethanol affect the cultivation? If it was 70% ethanol, so all Trypanosomes (especially in the blood) should be dead.

Lines 169-171. After the mosquitoes blood-feeding experiment, the authors separated only fed Cx. p. molestus because of autogeny, in other mosquito species fed specimens were recognized during dissection (presence of eggs). Why authors didn't separate all mosquito species after feeding? Because usually fed females are easy to distinguish and to separate them immediately would facilitate further work.

On what day after blood feeding were mosquitoes dissected? Please add this information.

Lines 163-167. The authors did not specify the temperature and humidity at which the colonies were maintained before the experiment. Please specify.

Author Response

Line 108-109. Authors wrote that collected keds and blood were stored in ethanol and to detect T. theileri they used cultivation. What percentage of ethanol was used? And can material fixation in ethanol affect the cultivation? If it was 70% ethanol, so all Trypanosomes (especially in the blood) should be dead.

                The reviewer is absolutely right that ethanol would kill trypanosomes. Collected keds were stored alive in zip-lock bags, only after their death they were fixed in ethanol. Game blood was processed using two methods: a part was fixed in ethanol, and a part was inoculated into vials with a medium for cultivation.

Lines 169-171. After the mosquitoes blood-feeding experiment, the authors separated only fed Cx. p. molestus because of autogeny, in other mosquito species fed specimens were recognized during dissection (presence of eggs). Why authors didn't separate all mosquito species after feeding? Because usually fed females are easy to distinguish and to separate them immediately would facilitate further work.

Almost all of Cx. p. quinquefasciatus mosquito females were fed after experiments, so we decided not to manipulate with the freshly fed females; the presence of eggs was used as a secondary control of bloodfeeding.

On what day after blood feeding were mosquitoes dissected? Please add this information.

                All dissections were done after defaecation, which means after day 10 PI for mosquitoes, and day 7 for sandflies, resp.     

Lines 163-167. The authors did not specify the temperature and humidity at which the colonies were maintained before the experiment. Please specify.

                Colonies are maintained at 25° C and 80% relative humidity.

We have corrected the missing data according to the reviewer´s suggestions.

Reviewer 3 Report

This well presented work requires only the slightest changes.  You cover the question of vector competence and hence impotance in transmission thoroughly in your paper, and in the light of this I suggest a slight change of title.  You might, for instance, use the term "new potential vectors". 

I have used comment and mark-up boxes in the attached version of your MS to point out some trivial infelicities of syntax, and a slight scientific query, which you may wish to attend to.

Author Response

The manuscript has been revised according to suggestions.